# Visual Relationship-Based Identification of Key Construction Scenes on Highway Bridges

**Chen Wang, Jingguo Lv *, Yu Geng and Yiting Liu**

School of Geomatics and Urban Spatial Informatics, Beijing University of Civil Engineering and Architecture, Beijing 102612, China; 2108160220002@stu.bucea.edu.cn (C.W.); 201904020125@stu.bucea.edu.cn (Y.G.); 202003020130@stu.bucea.edu.cn (Y.L.)

* Correspondence: lvjingguo@bucea.edu.cn

**Abstract:** Highway bridges play an important role in traffic construction; however, accidents caused by bridge construction occur frequently, resulting in significant loss of life and property. The identification of bridge construction scenes not only keeps track of the construction progress, but also enables real-time monitoring of the construction process and the timely detection of safety hazards. This paper proposes a deep learning method in artificial intelligence (AI) for identifying key construction scenes of highway bridges based on visual relationships. First, based on the analysis of bridge construction characteristics and construction process, five key construction scenes are selected. Then, by studying the underlying features of the five scenes, a construction scene identification feature information table is built, and construction scene identification rules are formulated. Afterward, a bridge key construction scene identification model (CSIN) is built; this model comprises target detection, visual relationship extraction, semantic conversion, scene information fusion, and identification results output. Finally, the effectiveness of the proposed method is verified experimentally. The results show that the proposed method can effectively identify key construction scenes for highway bridges with an accuracy rate of 94%, and enable the remote intelligent monitoring of highway bridge construction processes to ensure that projects are carried out safely.

**Keywords:** construction scene identification; visual relationship detection; scene rules; deep learning; neural networks; highway bridges



## 1. Introduction

Highway bridge construction is an important element of road transport, and plays an increasingly significant role in the development of the transportation sector. In the actual bridge construction process, the complex operating on-site environments, large numbers of construction personnel, and irregular operation of equipment often lead to major safety accidents [1], resulting in significant life and economic losses to societies and families [2]. Therefore, the identification of workers, equipment, and the behavioral relationship between workers and equipment at bridge construction sites, and thus, the inference of the current construction scene, has important application value for construction safety prevention.

The earliest methods used for construction safety monitoring relied primarily on manual monitoring during construction and safety assessment after completion [3]. However, owing to factors such as a wide working area, the large number of people on the construction site, and the complexity of the equipment used, a reliance only on manual point-to-point monitoring is often time-consuming and labor-intensive, and the monitoring results are prone to error.

Most current researchers use deep learning methods in artificial intelligence (AI) for safety monitoring during construction processes [4], with a focus on target detection of construction workers wearing helmets and holding equipment [5]. However, this method

ignores the interrelationship between workers and construction objects, leading to a lack of early warning capability for safety monitoring when workers perform non-compliant construction operations.

In recent years, more researchers have focused on visual relationship detection in deep learning, which aims to determine the topological relationship between targets in a scene [6,7] and generate the triplet form of subject–predicate–object. This approach can more accurately represent and describe construction scene information and contextual relationships. R-CNN [8] was used by VDR [9] to obtain the target candidate frame, and the relationship likelihood score of the triplet was obtained by a visual model and a semantic model for relationship prediction. Different from VRD, VTransE [10] was an end-to-end model that maps the visual features of targets into a low-dimensional relational space, using transfer vectors to represent the relationships between targets. The textual representation of subject/object was used by CAI [11] as contextual information to establish a visual relationship detection model. Features are the basis of target identification, so more features are incorporated into the DR-Net model to count the occurrence probability of subjects, predicates, and objects by visual features, spatial structure features, and relational features [12]. In order to better understand the relationship between targets, ViP-CNN [13] was used to establish the association between subjects, predicates, and objects on visual features by passing information between different models at the same layer. Zoom-net [14] was used for deep information transfer between local target features and global predicate relation features to the achieve deep integration of subjects and predicates. At present, visual relationship detection has been applied to a variety of image understanding tasks, such as image understanding in construction scenes. Wu et al. [5] performed relationship detection between workers and equipment by obtaining the head pose and body orientation of the worker. Kim et al. [15] reconstructed individual behaviors using object types of interactions between workers and equipment to improve construction scene identification. Xiong et al. [16] applied visual relationship detection in construction to a video surveillance system, enabling further improvement with respect to the immediate effectiveness of construction safety warnings. The above methods are able to identify specific targets and interrelationships between targets in construction scenes, but fail to further realize scene identification and understanding on this basis, and thus cannot achieve automation and intelligence in safety monitoring during construction. In addition, owing to the relatively high complexity of construction scenes, it is easy to encounter the problem of missing and incorrect detection of targets.

Visual relationship detection fully presents all information in an image and solves the problem of object relationship fragmentation caused by using target detection algorithms alone. However, there are only a few applications of visual relationship detection in highway bridge construction. In order to achieve intelligent safety monitoring of the bridge construction process and to complete construction scene identification and understanding, this paper proposes a visual relationship-based method for construction scene identification on highway bridges. The method combined the construction characteristics of highway bridges, and is based on the idea of deep learning. In this method, scene identification rules are formulated according to the target features and interrelationships in the construction scenes, and a scene identification model is then built based on the rules to complete the textual output of key scene information. The main work of this paper is as follows:

(1) Selection of key construction scenes on bridges. There are numerous bridge construction processes. Therefore, in this study, five key construction scenes of a bridge were selected based on an analysis of its construction characteristics and construction process.

(2) Formulation of identification rules for key construction scenes on bridges. A feature is the basis of scene identification. This study examines the underlying features that can distinguish the categories of key construction scenes, and establishes a feature information table and a tree diagram for the identification of key construction scenes on highway bridges. On this basis, the identification rules under different construction scenes are formulated.

(3) Building an identification model for key construction scenarios on bridges. In the target detection module, a feature pyramid network (FPN) and color moments are introduced to perform the multiscale detection of targets and obtain construction personnel identity information, while reducing the rate of missing and incorrect detection of targets. In the visual relationship extraction module, feature vectors are introduced to connect subjects, objects, and predicates in construction scenes in order to determine the interaction relationship between targets. In the semantic conversion module, frequency baselines are introduced to count the number of predicates in the construction scene, and the probability distribution of construction personnel actions is then obtained. In the scene information fusion module, an image–text encoder is introduced to combine the image results with the detection results to obtain the correspondence between the images and text. In the scene identification results output module, a rule consistency matching strategy is introduced to match the detected feature results with the formulated rules, and the category information of key construction scenes of highway bridges is then obtained.

(4) Validation of scene identification method. Experimental validation was performed using a homemade key construction scene identification dataset on a highway bridge. In addition, the accuracy, precision, recall, and other evaluation indexes were used to evaluate the accuracy of the proposed scene identification method. Moreover, we performed a comparative analysis with other visual relationship-detection methods to prove the effectiveness of the proposed method.

## 2. Proposed Method

### 2.1. Selection of Key Construction Scenes on Bridges

#### 2.1.1. Analysis of Bridge Construction Characteristics

In highway bridge engineering, there is a degree of difference between its production and general industrial production, which includes the following three perspectives.

(1) Large span of engineering structures. Highway bridge projects are often used to connect two distant areas; therefore, the bridge body has a long span. Furthermore, gantry cranes are essential types of equipment for the transport and installation of bridge bodies, but are more dangerous.

(2) More open-air and high-altitude operations. The fixed nature of highway bridge locations makes construction workers often face open-air work and to work from heights. As the distance of construction workers from the ground increases, the risk factor also increases layer-by-layer.

(3) High periodicity and repetitiveness. Bridge projects involve the use of similar types of structures, the same part of the sub-section construction, as well as other factors during the construction process. Therefore, they need to be carried out in a step-by-step manner, such as embedding steel casing, fixed formwork installation, concrete pouring, etc., which gives the bridge construction a certain periodicity and repetitiveness.

Owing to the aforementioned characteristics of highway bridge construction, there are a number of difficulties and safety hazards. To reduce the occurrence of accidents, it is necessary to monitor the bridge construction scene in real time. However, the bridge construction process is complex and varied; therefore, five key scenes were selected for this study.

#### 2.1.2. Key Construction Scenes on Bridges

The construction process of a highway bridge consists mainly of in situ construction and assembly construction. That is, the formwork and stand are set up at the location of the entire bridge, followed by the welding of the reinforcement and concrete pouring. After the concrete reaches its target strength, the formwork and stand are removed. Finally, prefabrication of the beams and bridge deck construction is carried out near the bridge site. The flow of the construction process is illustrated in Figure 1.

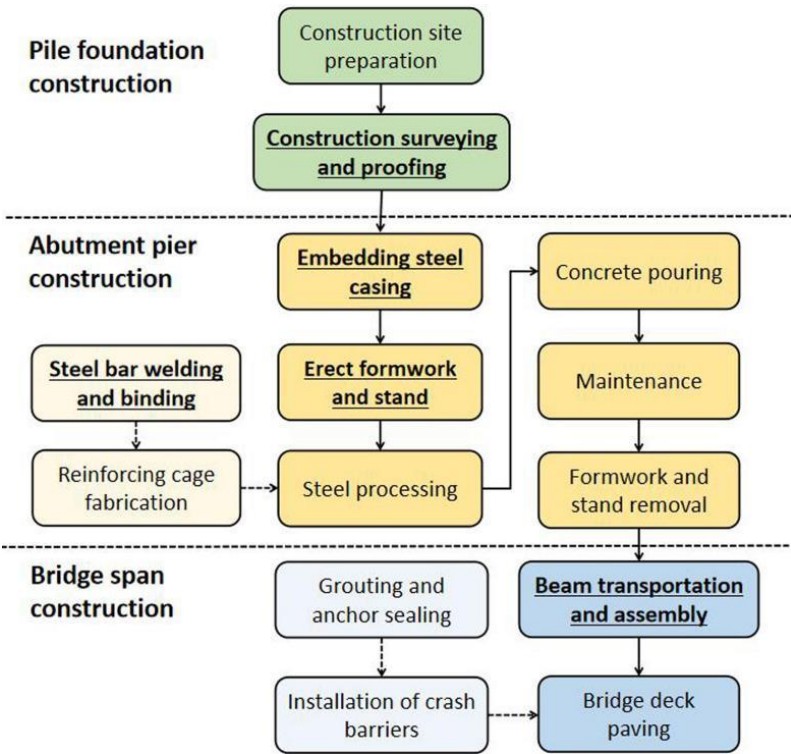

**Figure 1.** Schematic diagram of the bridge construction process.

Bridge construction is divided into three main stages: pile foundation construction, abutment pier construction, and bridge span structure construction, each of which is a complex and tedious construction process. In this study, five key construction scenes (indicated by bold underlining in Figure 1) were selected for analysis and identification.

(1) Construction surveying and proofing. Before the construction of the bridge project, the technicist should first carry out measurement lofting and data calculation on-site to provide the construction direction for the entire project. This is the premise and foundation for ensuring the quality of bridge projects.

(2) Embedding of steel casing. During bridge construction, to achieve the required load-bearing capacity, the steel casing needs to be embedded to ensure the verticality of the bridge and prevent collapse caused by the falling of debris around it. This directly affects the stability of the bridge pile foundations.

(3) Erection of formwork and stand. When pouring the superstructure of the bridge on-site, the first step is to erect a stand at the location of the bridge hole to support the formwork and poured reinforced concrete. This is an important construction step in bridge engineering.

(4) Steel bar welding and binding. Steel processing is an extremely important step in bridge construction, and the welding and binding of steel bars are basic links in steel processing to ensure the stability of steel installation. This, in turn, affects the structural safety of the entire bridge.

(5) Beam transportation and assembly. The weight and volume of the equipment involved in the beam transportation and assembly stages are large, such as gantry cranes and bridge erectors, which are prone to accidents if not operated carefully. This is a major source of danger during the bridge construction process.

To ensure the stability and safety of bridge structures, it is necessary to strengthen the management of the construction process, particularly during the key construction scenes. The first step in management is to accurately identify the current scene information and monitor hazards according to the interrelationship between workers

and equipment. Based on this idea, this study proposes identification rules for key construction scenes on bridges.

## 2.2. Formulation of Identification Rules for Key Construction Scenes on Bridges

Scene identification can be achieved by extracting the underlying features of different instances in an image and the spatial location relationship between them, and by inferring the relationship to output, the current scene information of that image. Based on this idea, this study designed identification rules for key construction scenes on highway bridges. Table 1 presents the information of bridge key construction scenes identification features: Table 1 A presents the key-scene construction equipment information, and Table 1 B presents the key-scene construction personnel and construction material information. In both tables, the underlying features required to identify the five key scenes are marked as "√". Where, ①–⑤ denote five key construction scenes on the bridge. ① denotes construction surveying and proofing, ② denotes embedding steel casing, ③ denotes erect formwork and stand, ④ denotes steel bar welding and binding, and ⑤ denotes beam transportation and assembly.

Figure 2 shows the rules of bridge key construction scene identification. It describes the logical relationship of the underlying features of construction personnel (blue), construction equipment (green), and construction materials (orange). The left shows the five key construction scenes and the corresponding construction equipment for each scene.

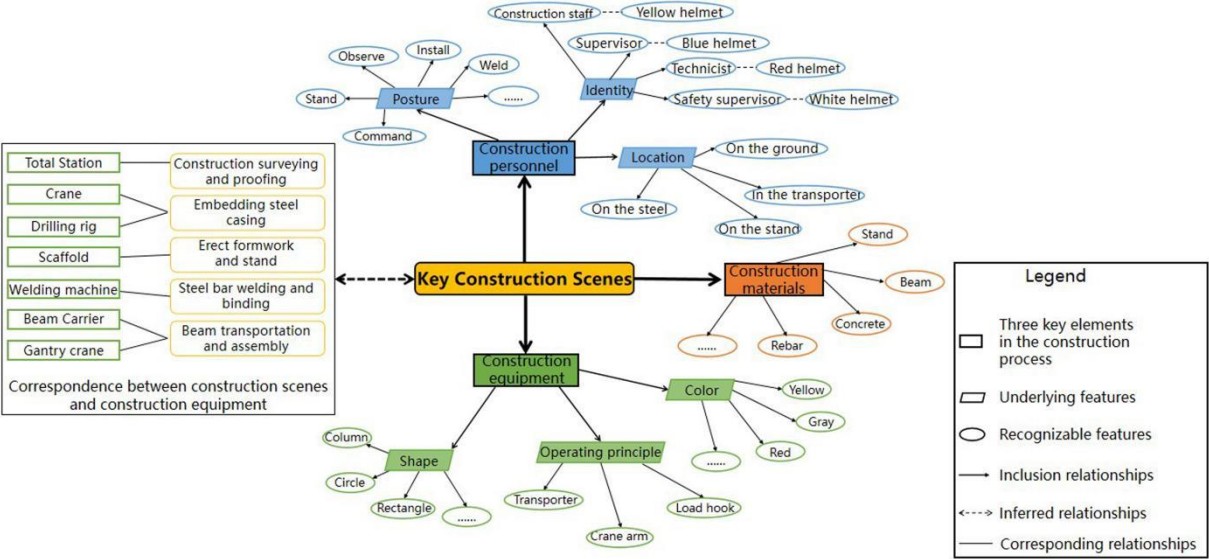

**Figure 2.** Identification rules tree diagram of bridge key construction scenes.

(1) Construction personnel include three underlying features: posture, identity, and location. ① Posture features include seven kinds: observe, command, stand, etc.; ② Identity features include construction stuff (yellow helmet), supervisor (blue helmet), etc.; and ③ Location features include four types of location information such as on the ground, on the stand, etc.

(2) Construction equipment includes three underlying characteristics of shape, color, and working principle. ① Shape characteristics include cylindrical, round, etc.; ② Color characteristics include red, yellow, etc.; ③ Working principles include arm, hook, etc.

(3) Construction materials include rebar, concrete, etc.

**Table 1.** (**A**) Construction equipment in five scenes; (**B**) Construction personnel and construction materials in five scenes.

**(A)**

**Construction Equipment**

| Scenes | Name | | | | | | | Shape | | | Color | | | Principle | | |
|---|---|---|---|---|---|---|---|---|---|---|---|---|---|---|---|---|
| | Total Station | Crane | Drilling Rig | Scaffold | Welding Machine | Beam Carrier | Gantry Crane | Column | Circle | Rectangle | Red | Yellow | Gray | Transporter | Crane Arm | Load Hook |
| ① | √ | | | | | | | | | √ | | | | | | |
| ② | | √ | √ | | | | | √ | | | | | | | | |
| ③ | | | | √ | | | | | | | | | | | | √ |
| ④ | | | | | √ | | | √ | | | | | √ | | | |
| ⑤ | | | | | | √ | √ | | | | | | | √ | √ | |

**(B)**

**Construction Personnel** / **Construction Materials**

| Scenes | Posture | | | | | | Identity | | | | Location | | | | Construction Materials | | | |
|---|---|---|---|---|---|---|---|---|---|---|---|---|---|---|---|---|---|---|
| | Observe | Command | Stand | Install | Weld | Transport | Yellow Helmet | Blue Helmet | Red Helmet | White Helmet | On the Ground | In the Transporter | On the Stand | On the Steel | Rebar | Stand | Beam | Concrete |
| ① | √ | | | | | | | √ | √ | | √ | | | | | | | |
| ② | | √ | | | | | √ | | | | √ | | | | | | | |
| ③ | | | √ | √ | | | √ | | | | | | √ | | | | √ | |
| ④ | | | | | √ | | √ | | | | | | | | √ | √ | | |
| ⑤ | | | | | | √ | | | | √ | | | √ | | | | | √ |

For example, the worker wearing a blue helmet stands on the ground observing the total station and instructing the worker wearing a yellow helmet, it can be inferred that the current scene is "construction surveying and proofing"; the worker wearing a yellow helmet holds a cutting machine to weld long objects, and it can be inferred that the current scene is "steel bar welding and binding".

### 2.3. Building of Identification Model for Key Construction Scenes on Bridges

The authors in [17] proposed a relationship detection model named RelDN, and this study draws on the idea of constructing a CSIN network model with CNN (Convolutional Neural Network) [18] and DCR (Deep Convolutional Relationship) [19] as the basic framework for bridge construction scene identification. The structure of CSIN is shown in Figure 3. There are four parts. The first part is image input and feature extraction part, which is composed of the convolutional neural network CNN to extract the underlying features of construction images. The second part is the feature processing part, which mainly includes the target detection module, visual relationship extraction module, and semantic conversion module to obtain different feature score charts. The third part is the feature fusion part, which is composed of the scene information fusion module to fuse image features and text features. The fourth part is the result output part, which is composed of the scene identification result output module to obtain the current construction scene information.

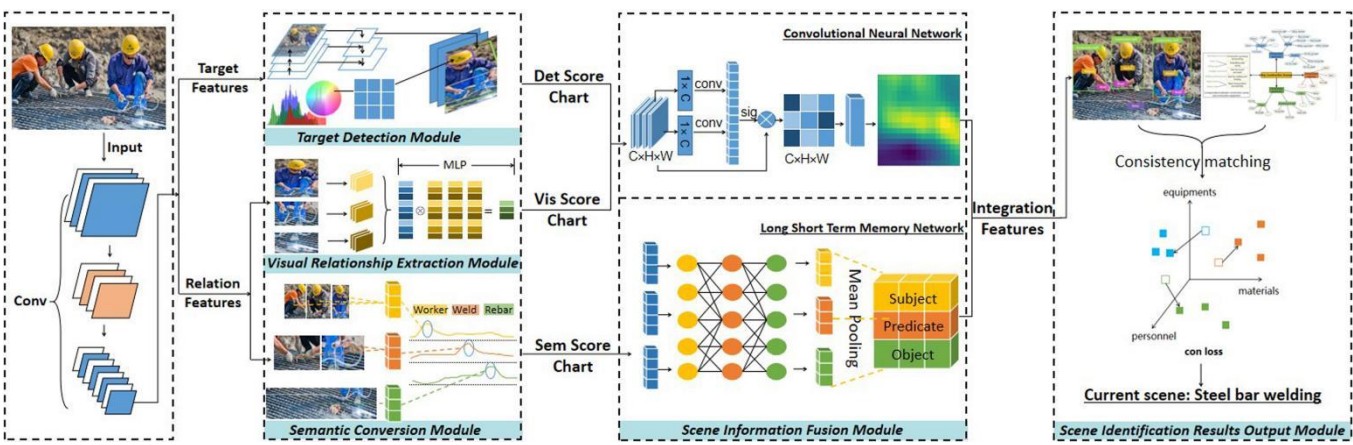

**Figure 3.** Structure schematic diagram of bridge construction scene identification model. (The source of the identifiable image in Figure 3 is shown in the Supplementary Materials).

#### 2.3.1. FPN-Based Target Detection Module

Detecting and locating various types of targets in construction images are the basis for achieving construction scene identification; therefore, this study first needs to extract and capture feature information, such as the location and category of construction personnel, using the target detection module. To address the problem of target size difference and target miss detection in construction scene images owing to the imaging angle, a detection method that can cope with such multi-scale variation is needed. The feature pyramid network (FPN) [20] can feature extraction for each scale of the image, increasing the perceptual field of the bottom layer of the feature map. So, the FPN is able to obtain more contextual information when performing small target detection at the bottom layer, reducing the rate of missing and incorrect detection. Therefore, this study adds an FPN in the target detection module, which makes shallow networks focus more on detailed information, and high-level networks focus more on semantic information.

In addition, standardized coordinates were used to encode the bounding box between targets to obtain position information and to complete the prediction of the position relationship.

$$\Delta b_1, b_2 = \left(\frac{x_1 - x_2}{W_2}, \frac{y_1 - y_2}{H_2}, \log\frac{W_1}{W_2}, \log\frac{H_1}{H_2}\right) \tag{1}$$

$$c(b) = \left(\frac{x}{W_{img}}, \frac{y}{H_{img}}, \frac{x + W}{W_{img}}, \frac{y + H}{H_{img}}, \frac{WH}{W_{img}H_{img}}\right) \tag{2}$$

where $b_1, b_2$ are the bounding boxes between the two targets, $\Delta b_1, b_2$ denote the increments between the bounding box coordinates, and $(x, y, W, H)$ is the coordinate information of the bounding box. In addition, $c(b)$ denotes the normalized coordinate feature of the bounding box, and $W_{img}, H_{img}$ are the width and height of the input image, respectively.

To address the problem of mismatch between helmet type and construction personnel identity, in this paper, the color characteristics of the safety helmet are extracted using the color moment method [21]. The first-order moments describe the average color of the safety helmet, the second-order moments describe the color variance, and the third-order moments describe the offset of the color. Thus, the color moments can present comprehensive color characteristics of the safety helmet to achieve the purpose of corresponding with the identity of workers. The correspondence between helmet color and worker identity is shown in Table 2. The formulae for calculating the first-order, second-order, and third-order moments are as follows:

$$M_1 = \frac{1}{N}\sum_{j=1}^{N} P_{ij} \tag{3}$$

$$M_2 = \left(\frac{1}{N}\sum_{j=1}^{N}(P_{ij} - M_1)^2\right)^{\frac{1}{2}} \tag{4}$$

$$M_3 = \left(\frac{1}{N}\sum_{j=1}^{N}(P_{ij} - M_1)^3\right)^{\frac{1}{3}} \tag{5}$$

**Table 2.** Matching relationship between safety helmet color and construction personnel identity.

| Color Classification | Red Helmet | Yellow Helmet | Blue Helmet | White Helmet |
|---|---|---|---|---|
| Worker status | Technicist | Construction staff | Supervisor | Safety supervisor |

2.3.2. Visual Relationship Extraction Module Based on Feature Vectors

The visual relationship detection branch is used to capture deeper visual features in construction images, including the construction scene content, interrelationships, and logical relationships between different objects. The visual relationship extraction module focuses on obtaining the interaction probability values between the construction action sender (subject), construction action (predicate), and construction object (object), as shown in Figure 2. This module generates a set of class vector logits conditioned on region-of-interest (ROI) feature maps and passes the fused feature map information so that the network can fully learn and perceive the visual and semantic intersection information in the construction scene. A multilayer perceptron (MLP) is used to connect the feature vectors of the subject, predicate, and object to obtain the probability values of the interaction relationships between different entity targets in the construction scene. The formula is as follows:

$$f(x) = G(b^{(2)} + W^{(2)}(s(b^{(1)} + W^{(1)}x))) \tag{6}$$

where W is the connection weight, b is the bias, G is the softmax function, and s is a sigmoid function.

To improve the processing efficiency of the network for visual information and reduce the computational cost of the network, two cross-layer connections [22] are constructed in the visual relation extraction module. Then the subject/object ROI features extracted by the detection module are mapped to the predicate class vector logits to facilitate the transfer and flow of information in the network.

### 2.3.3. Semantic Conversion Module Based on Frequency Baseline

A construction scene graph contains not only intuitive visual information, but also deep semantic information. Scene graphs are one of the methods used to construct the visual relations of images [23]. The main idea is to divide the visual relations between all objects in an image into a triadic subject–predicate–object form, which is used as a whole learning task [24,25]. The semantic conversion module focuses on outputting the relationship information between the subject and object. This module draws on the idea of a scene graph to generate a set of binary relational feature maps of the ROI and passes the semantic information extracted by the relationship detection branch to a higher level of cyberspace. The interrelationships and attribute information between different objects are then captured by calculating the frequency of predicates between subjects and objects. This predicate is generally limited and regular; for example, the relationship between construction workers and scaffolding is generally workers "install" scaffolding or workers "stand" on scaffolding, but not other predicates such as "wear". Therefore, to improve the processing and learning efficiency of the semantic conversion model, a frequency baseline was set based on the number of occurrences of the predicate [26]. For any pair of training images, the prediction probability distribution was obtained by counting the number of occurrences of subject s and object o in the real box with the set frequency baseline.

$$\omega(s, o) = 1 - p(pred = \varnothing | s, o) \tag{7}$$

where $p(pred|s, o)$ denotes the probability of predicate distribution between subject s and object o, and $p(pred = \varnothing | s, o)$ denotes that there is no interrelationship between subject s and object o.

To prevent the network from incorrectly inferring two targets that are close but not interrelated, a loss function L is designed when subject s and object o are interrelated to maximize the bounding box distance between the two targets determined by the predicate.

$$\begin{aligned} L = &\frac{1}{N} \sum_{i=1}^{N} \frac{1}{\left|P\left(O_i^+\right)\right|} \sum_{p \in P\left(O_i^+\right)} \max(0, \alpha - m^s(i, p)) \\ &+ \frac{1}{N} \sum_{j=1}^{N} \frac{1}{\left|P\left(O_j^+\right)\right|} \sum_{p \in P\left(O_j^+\right)} \max(0, \alpha - m^o(j, p)) \end{aligned} \tag{8}$$

where $P()$ is the specific set of predicates associated with the input, p represents the predicate class, and $O_i^+, O_j^+$ denote the set of targets whose relationship is p. In addition, $\alpha$ is the threshold value, $m^s, m^o$ denotes the confidence of the subject and object, and i, j denotes the index of the subject and object.

### 2.3.4. Scene Information Fusion Module Based on Image-Text Encoder

After the target detection module and visual relationship extraction module, the image information of the construction personnel and the image information of the subject and object in the scene were obtained. Moreover, the text information of the predicate in the scene was obtained after the semantic conversion module. The key step in realizing scene identification is to combine image information with text information. In this study, the scene information fusion module was formulated by referring to the method of correlation description between images and text in the literature [27].

First, the detection, visual, and semantic scores obtained by the three modules are softmax normalized to obtain the target relationship probability $P^{pre}$.

$$P^{pre} = \text{softmax}(f_{Det} + f_{Vis} + f_{Sem}) \tag{9}$$

where $f_{Det}$, $f_{Vis}$, and $f_{Sem}$ denote the output relationship probabilities of the target detection module, visual relationship extraction module, and semantic conversion module, respectively.

After obtaining the target relationship probabilities, the output image results and detection results are encoded to the same dimension by the image encoder $\phi$ through convolutional neural networks (CNNs) [28], and the text encoder $\varphi$ through the long short-term memory network (LSTM) [29]. Then, the cosine similarity between the paired image results and the detection results was calculated to construct the ranking loss function. The ranking loss function of encoder $L_{rank}$ is as shown in Equation (10).

$$\begin{aligned} L_{rank} = \min_{\theta} \sum_x \sum_k \max\{0, \alpha - s(\phi(x), \varphi(t)) + s(\phi(x), \varphi(t_k))\} \\ + \sum_t \sum_k \max\{0, \alpha - s(\phi(x), \varphi(t)) + s(\phi(x_k), \varphi(t))\} \end{aligned} \tag{10}$$

where $\theta$ denotes all parameters in the image encoder and text encoder, $\alpha$ is the boundary value, and s is used to calculate the cosine similarity between the image embedding vector $\phi(x)$ and the detection result embedding vector $\varphi(t)$; $x_k$, $t_k$ denote the mismatched images and texts, respectively.

### 2.3.5. Scene Identification Results Output Module Based on Rule Consistency Matching

The four modules above are all intermediate results, which can be expressed as "features," while the final goal of this study is to output a textual expression that is consistent with the scene image to be detected. The textual output of scene identification is obtained by matching the integration features acquired from the scene information fusion module with scene identification rules (Figure 2). In this paper, the method of reference [30] is referred to, and the loss function $L_{con}$ is used to calculate the consistency between the integration features and the rules. $L_{con}$ is calculated as shown in Equation (11).

$$L_{con} = \left(\frac{1}{a}\sum_i^a \frac{u_i^T x_i^t}{||u_i|| \cdot ||x_i^t||} + \frac{1}{b}\sum_j^b \frac{v_i^T y_i^t}{||v_i|| \cdot ||y_i^t||} + \frac{1}{c}\sum_j^c \frac{w_i^T z_i^t}{||w_i|| \cdot ||z_i^t||}\right)^2 \tag{11}$$

where $(u_i, x_i^t)$, $(v_i, y_i^t)$, and $(w_i, z_i^t)$ appear in pairs and denote subject-construction personnel matching, object-construction equipment matching, and predicate-posture matching, respectively; and a, b, and c represent the number of instances of the three construction elements (mentioned in Figure 2).

In the training process, given a dataset $D = \left\{(I_k, S_k)_{k=1}^N\right\}$ containing N image-text, a batch of images is sampled from the dataset for training, and the final loss function is a weighted sum L of the ranking loss and the consistency loss.

$$L = \sum_k^{N_b} L_{rank}(I_k, S_k) + \lambda_{con} \sum_k^{N_b} L_{con}(I_k, S_k) \tag{12}$$

where I denotes the image, S denotes the text, and $\lambda_{con}$ is a hyperparameter with an adjustable balance.

### 2.3.6. Method Flow-Chart

The overall method flow-chart is shown in Figure 4. Firstly, the bridge construction scene images are input into the convolutional neural network, then the geometric features and color features in the shallow layer of the image are extracted by the operations of convolution, pooling, and full connection to form the feature maps. Then the extracted

feature maps are fed into the target detection module, the visual relationship extraction module, and the semantic conversion module through the target detection branch and the relation detection branch, respectively, to obtain information on parameters such as location, category, probability values, and attributes of image targets to form a detection score chart, visual score chart, and semantic score chart. Afterward, the three score charts are fed into the convolutional neural network and the long short-term memory network, respectively, for image coding and text coding to obtain the integration features. Finally, the integration features are matched with the scene identification rules for consistency, and the current scene information is obtained and output in the text form.

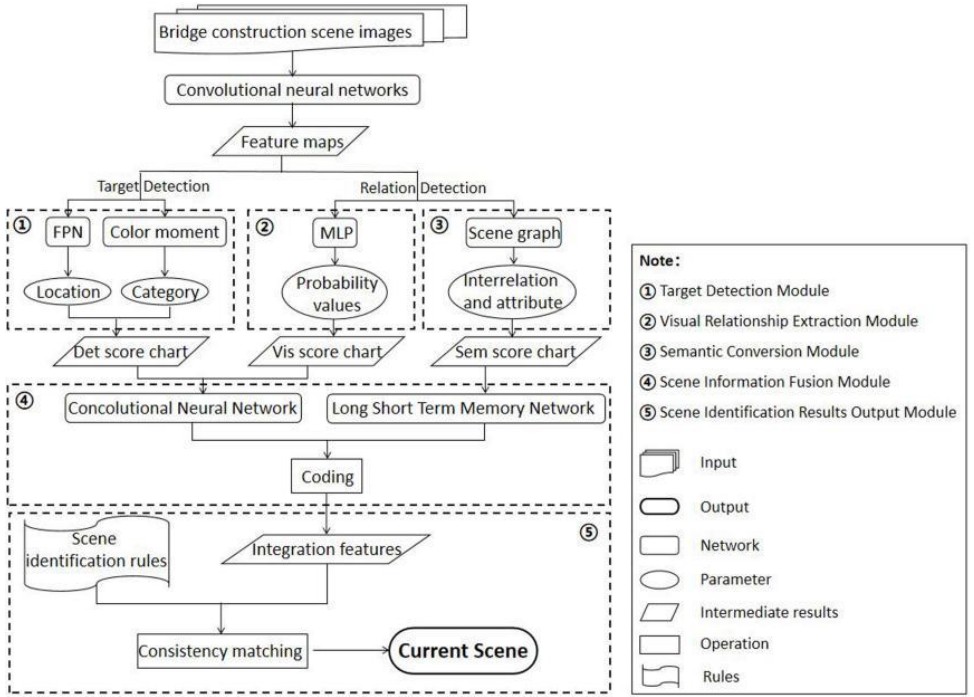

**Figure 4.** Method flow-chart.

## 3. Experiment

### 3.1. Experimental Configuration

3.1.1. Dataset

To fully learn the target features, semantic features, and visual features in different construction scenes, and to identify key construction scenes on bridges, a large amount of data is required for training. Because there is no dataset for construction scene identification that satisfies the needs of this study, a scene-based construction identification dataset for highway bridges is built in this study. For the five key scenes mentioned in Section 2.1.2, the construction scene images are intercepted by online bridge construction monitoring videos considering various factors such as the target size variation, location distribution, and similar color interference. In addition, LabelImg is used to label visual information such as the location and category of targets, as well as the semantic information of the interrelationship between targets in the images. So, the model can fully learn and understand the logical relationships embedded in the images. The specific information of the bridge key construction scene identification dataset constructed in this study is presented in Table 3, containing a total of 465 images. This dataset was constructed from three aspects: subject, object, and predicate. Furthermore, 60% of the images were selected as the training set and the remaining 40% were selected as the test set, including 37 images for each of the five key construction scenes.

**Table 3.** Number and interrelationship of images in each construction scene.

| Bridge Construction Scene | Number of Images | Visual Relationship | | |
|---|---|---|---|---|
| | | Subject | Predicate | Object |
| Construction surveying and proofing | 95 | Worker | Observe | Total station |
| Embedding steel casing | 90 | Crane | Conduct | Steel case |
| Erect formwork and stand | 90 | Worker | Install | Scaffold |
| Steel bar welding and binding | 100 | Worker | Weld | Rebar |
| Beam transportation and assembly | 90 | Beam carrier | Transport | Beam |

3.1.2. Evaluation Indicators

To verify the accuracy of the proposed bridge key construction scene identification method, indicators such as accuracy (Acc), precision (P), and recall @K (R@K) were used to evaluate the results of the experiments. The main formulae are shown in Equations (13)–(15).

$$\text{Accuracy} = \frac{\text{TP} + \text{TN}}{\text{TP} + \text{TN} + \text{FP} + \text{FN}} \tag{13}$$

$$\text{Precision} = \frac{\text{TP}}{\text{TP} + \text{FP}} \tag{14}$$

$$\text{Recall@K} = \frac{\text{TP@K}}{(\text{TP@K}) + (\text{FN@K})} \tag{15}$$

Among them, true positive (TP) and true negative (TN) are correct detection results, false positive (FP) is wrong detection, and false negative (FN) is missed detection.

3.1.3. Implementation Details

The configuration of this experimental platform is the Windows 10 operating system and CUDA 10.1 computing platform; the algorithm framework is TensorFlow-GPU1.12.0 and Keras2.13; the programming language is Python 3.6.13. To obtain a better training effect, the size of the image input network was set to 800 pixels in the training phase. Then, the batch was set to 1, the number of iterations was 10,000, and the initial learning rate was 0.001.

*3.2. Identification Results and Accuracy Analysis for Key Construction Scenes on Bridges*

To verify the effectiveness of the bridge key construction scene identification method proposed in this study, two parameters, namely the identification effect and identification accuracy, were evaluated and analyzed.

3.2.1. Scene Identification Results and Analysis

To verify the scene identification effect of the proposed method, experiments were conducted on the test set. Figure 5 shows some of the data in the test set, including five key construction scenes: (a) shows three technicists wearing red helmets to operate the total station and recording; (b) shows that the steel case is controlled by the crane arm, and the crane is operated by two construction workers wearing yellow helmets; (c) shows three construction personnel in yellow helmets welding steel bars with electric welders; (d) shows two construction workers in yellow helmets standing on the support to install the scaffold; and (e) shows a construction worker in a yellow helmet directing the beam transporter to transport the beam.

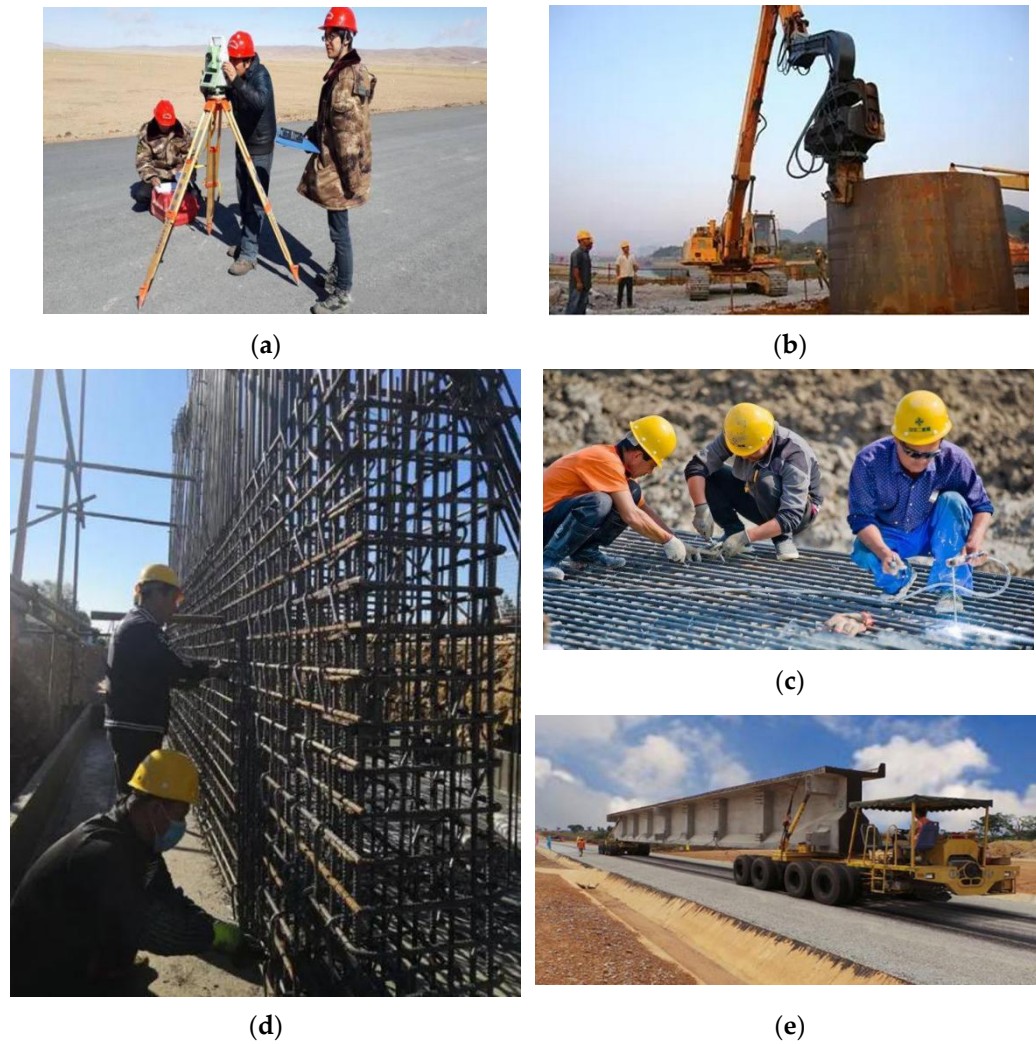

**Figure 5.** Partial data in the test set (five key construction scenes). (The sources of the identifiable images in Figure 4 are shown in the Supplementary Materials.)

The scene identification method proposed in this paper was applied to the test set for the experiment. Figure 6 and Table 4 show the scene identification results.

By analyzing the identification results in Figure 6 and the information in Table 4, it was found that the proposed scene identification method can correctly output the final scene category information (black box in the upper left corner in Figure 6). The information is derived from the intermediate results by reasoning through the formulated scene identification rules. The intermediate results consisted of two parts: the target detection result (green) and the visual relationship detection result (yellow for the subject, purple for the object, and pink for the predicate).

From the target detection results, we can see that the proposed method can distinguish different identity types according to the color of the helmet worn by workers, such as the detection result for workers wearing helmets in Figure 6; (a) is a "technicist", while the workers wearing yellow helmets in (b–e) are detected as "construction staff". In particular, the method proposed in this paper can still accurately detect the type and location information of the relatively small-sized workers appearing on the left side of (e) (this result will be analyzed in Comparison Results and Analysis of Target Detection Module). The visual relationship detection results show that the proposed method can correctly identify the subject, predicate, and object, and can connect the above three through the red line segment to reflect the correlation between them. Finally, the final scene identification results were obtained from the above two intermediate results using inference rules.

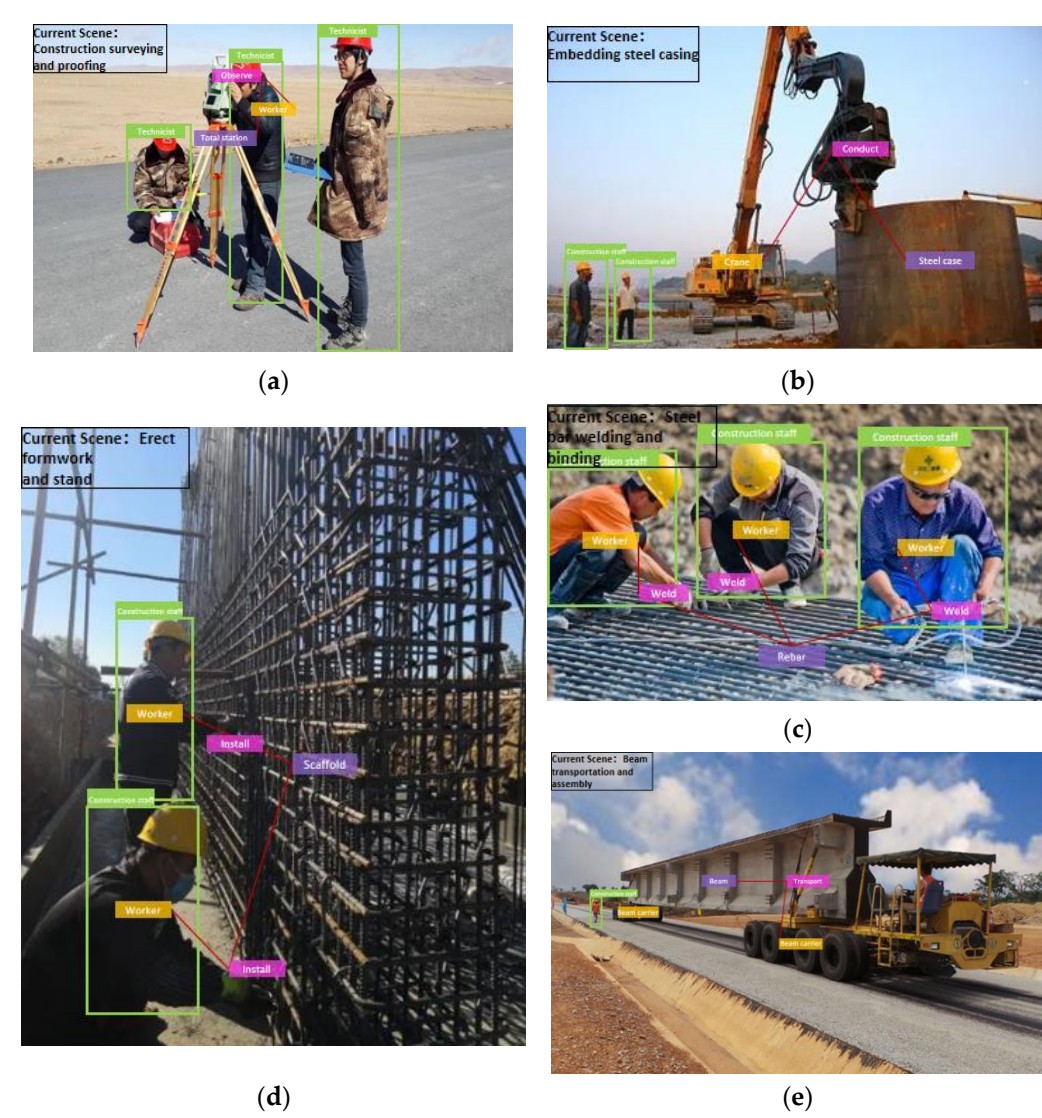

**Figure 6.** Partial results of the bridge key construction scene identification method on the test set. (The sources of the identifiable images in Figure 5 are shown in the Supplementary Materials.)

**Table 4.** Information table showing identification results for key construction scenes on bridges.

| Figure | Intermediate Results | | | | Final Results |
|---|---|---|---|---|---|
| | Target Detection Results | Visual Relationship Detection Results | | | Construction Scene Category Information |
| | | Subject | Predicate | Object | |
| A | Technicist | Worker | Observe | Total station | Construction surveying and proofing |
| B | Construction staff | Crane | Conduct | Steel case | Embedding steel casing |
| C | Construction staff | Worker | Weld | Rebar | Erect formwork and stand |
| D | Construction staff | Worker | Install | Stand | Steel bar welding and binding |
| E | Construction staff | Beam carrier | Transport | Beam | Beam transportation and assembly |

3.2.2. Scene Identification Accuracy and Analysis

To verify the scene identification accuracy of the method in this study, it was evaluated using a confusion matrix, as shown in Table 5. Each scene category contains 37 test images.

**Table 5.** Table of identification accuracy results for key construction scenes on bridges, where: ① denotes construction surveying and proofing, ② denotes embedding steel casing, ③ denotes erection of formwork and stand, ④ denotes steel bar welding and binding, and ⑤ denotes beam transportation and assembly.

|  |  | True Value | | | | | Precision (%) |
|---|---|---|---|---|---|---|---|
|  |  | ① | ② | ③ | ④ | ⑤ |  |
|  | ① | 37 |  |  |  |  | 100 |
| Predicted Value | ② |  | 35 |  |  |  | 100 |
|  | ③ |  |  | 31 | 3 |  | 91.2 |
|  | ④ |  |  | 6 | 34 |  | 85.0 |
|  | ⑤ |  | 2 |  |  | 37 | 94.9 |
| Recall (%) |  | 100 | 94.6 | 83.8 | 91.9 | 100 |  |
|  |  | Accuracy (%) = 94 | | | | | |

Comparing the data in the table, it can be seen that the identification accuracy and recall in the "construction surveying and proofing" are the highest, and 100% recognition can be achieved. The identification accuracy and recall in the "erection of formwork and stand" and "steel bar welding and binding" were lower, with accuracies of 91.2% and 85.0%, and recall values of 83.8% and 91.9%, respectively. From an analysis of the reasons, we found that the processes of "erect formwork and stand" and "steel bar welding and binding" have high similarity. It is obvious from (c,d) of Figure 5 that the above two scenes have confusing targets, so the identification accuracy is slightly lower than that of the other scenes. The identification accuracy of "beam transportation and assembly" is higher because the size of the beam transporter is larger than the targets in other scenes, which is easy to identify. The identification accuracy of the "construction surveying and proofing" is the highest because the target in this scene is clear and the background is simple, which is not easily disturbed by other information. However, in general, the scene identification accuracy of this study reached 94%, which can complete the identification of key construction scenes on bridges.

Based on the experimental results obtained, it can be concluded that the key construction scene identification method proposed in this paper has a good scene understanding ability. This method can fully learn the semantic and visual information in the graph, perform target localization and relationship detection, and accurately output the category information of the scene.

*3.3. Experimental Results and Analysis of Identification Model CSIN for Key Construction Scenes on Bridges*

The CSIN model proposed in this study plays an important role in the scene identification process. To verify its effectiveness, two aspects of the model, namely the overall performance and internal modules, were evaluated and analyzed.

3.3.1. Experimental Results and Analysis of the Performance for the Scene Identification Model

The PR curves were plotted using precision and recall, which can visually describe the model performance. Figure 7 shows the PR curves generated when the IoU threshold is set to 0.5, 0.6, and 0.7, where the horizontal and vertical coordinates represent recall and precision, respectively. From an analysis of the three curves in the figure, it can be seen that when the IoU threshold is set to 0.5, the PR curve is closer to the upper right; that is, the precision and recall are both higher. The area formed by the PR curve and coordinate axis gradually decreased as the IoU threshold increased. When the IoU is 0.6 and 0.5, the two PR curves start to decrease significantly at recall >0.7, and when the IoU is 0.7, the PR curves start to decrease around recall =0.5. This indicates that the proposed CSIN model had the best detection effect when the IoU threshold was 0.5.

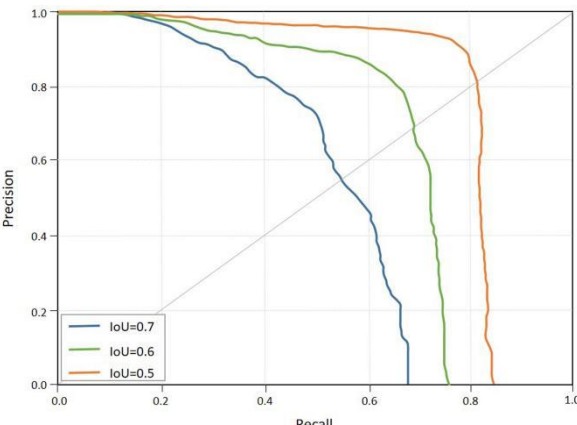

**Figure 7.** PR curves under different IoU thresholds.

3.3.2. Comparative Experimental Results and Analysis of Specific Modules

The CSIN model proposed in this study includes two important modules: target detection and visual relationship extraction. To verify the effectiveness of the proposed algorithms in these two modules, comparative experiments were conducted separately.

Comparison Results and Analysis of Target Detection Module

The basis of achieving scene identification is to correctly detect the classification and location information of targets in construction scenes; therefore, the effect of adding FPN in the detection module was tested in this study.

As shown in Figure 8, the second column is the true value, which is the feature map obtained from the original image after the grayscale comparison operation, and it is used to highlight specific regions of the foreground targets of the image. Further, the green box is the target with a smaller size. The third column is the convolutional heat map, which is used to delineate the target regions. The last two columns are the feature visualizations obtained by the two methods after channel-dimension averaging. Based on the results, the images (e,j) obtained by our method are generally clearer than the images (d,i) without FPN. In addition, both sets of feature maps contain targets of smaller size (red and yellow boxes), where the red boxes are marked by the multi-scale target detection without an FPN, and their response value is low when compared with the real value of the green boxes; that is, there is a missed detection. It is evident from the yellow boxes that the response value of the features is higher after the FPN is applied, and the human shape can be roughly detected. It can be concluded that the CSIN model proposed in this study, which applies an FPN for multi-scale target detection, is effective.

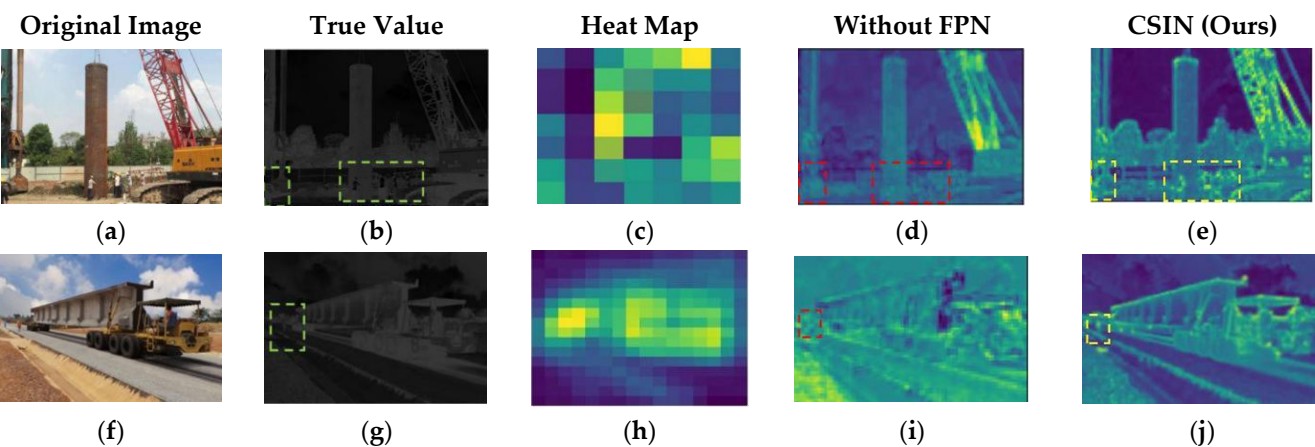

**Figure 8.** Visualization of the convolutional feature map obtained by channel-dimension averaging. (The sources of the identifiable images in Figure 5 are shown in the Supplementary Materials.)

Comparison Results and Analysis of the Visual Relationship Detection Module

Visual relationship detection is a prerequisite for achieving deep perception and understanding of construction scenes, and the degree of detection accuracy determines the merit of the CSIN. Therefore, in this paper, four mainstream algorithms are selected for visual relationship detection in the visual relationship extraction module for comparison experiments. Among them, VRD [9] is one of the earliest algorithms used for visual relationship detection and often appears as a comparison model; both Large-Scale [31] and the proposed method adopt the semantic module and visual module for feature extraction in model design; Motifs [26], Graph R-CNN [32], and the proposed method are all based on the basic model of scene graph for visual relationship detection. So, these four algorithms are selected for comprehensive comparison experiments. Their effectiveness can be assessed based on two aspects: subject/object localization accuracy and predicate detection accuracy, where subject/object localization focuses more on the target detection ability of the model, whereas predicate detection focuses more on the relationships.

Table 6 lists the visual relation detection results of the different algorithms. In general, the CSIN model in this study has better performance for the target detection of subjects and objects; for predicate detection, the CSIN model is not significantly different from other algorithms. Almost all of the models had the highest detection results for Recall@100.

**Table 6.** Comparison results with other visual relationship detection algorithms.

|  | Subject Detection | | | Object Detection | | | Predicate Detection | | |
|---|---|---|---|---|---|---|---|---|---|
| Recall at (%) | 20 | 50 | 100 | 20 | 50 | 100 | 20 | 50 | 100 |
| Large-Scale | 20.7 | 27.9 | 32.5 | 36.0 | 36.7 | 36.7 | 66.8 | 68.4 | 68.4 |
| Motifs | 21.4 | 27.2 | 30.3 | 32.9 | 35.8 | 36.5 | 58.5 | 65.2 | 67.1 |
| VRD | - | 0.3 | 0.5 | - | 11.8 | 14.1 | - | 27.9 | 35.0 |
| Graph R-CNN | - | 28.5 | 35.9 | - | 29.6 | 31.6 | - | 54.2 | 59.1 |
| CSIN (ours) | 35.9 | 37.8 | 42.4 | 36.1 | 36.7 | 37.0 | 65.3 | 67.9 | 69.3 |

Specifically, the subject detection accuracy in Recall@100 reached 42.4%, which is at least 6.5% or so higher than that of other algorithms. The object detection accuracy in Recall@100 reached 37.0%, which was also slightly higher than those of the other detection algorithms. However, the relationship detection was slightly lower than those of the other algorithms. Specifically, the predicate detection accuracies of Recall@20 and Recall@50 are 65.3% and 67.9%, respectively, which are lower than the predicate detection accuracy of the large-scale algorithm. This is because the large-scale algorithm is for the location and relationship detection of larger size targets. What is more, the predicate detection in this study does not show obvious superiority; this may be because bridge constructions are characterized by complex scenes and ambiguous relationships between workers and equipment, and it is relatively difficult to distinguish relationships. Subsequent experiments could be further improved to address this problem.

In summary, the localization accuracy results for the subject and object show that the use of an FPN can improve the detection accuracy of the target. The CSIN model proposed in this study works well for relationship detection and can effectively infer scene information during the construction process.

## 4. Discussion

In this part, we discuss three main points: robustness of scene identification rules, stability of the CSIN model detection frame, and generalization capabilities of the CSIN model.

### 4.1. Robustness of Bridge Construction Scene Identification Rules

The bridge construction scene identification rules developed in this paper adopt the idea of consistency matching. The logical relationship between construction personnel, construction equipment, and construction materials in the construction scenes is considered, which satisfies the needs of this paper to a certain extent. However, the lack of some reasoning strategies, such as inductive reasoning [33] and deductive reasoning [34], makes the constraint relationships among construction activities unable to be further refined into rules. BIM technology is used to obtain construction information [35,36] and obtain the constraint relationship between construction activities, so as to deduce the logical sequence between construction activities, which can improve the robustness of the scene identification rules to a certain extent.

### 4.2. Stability of the CSIN Model Detection Frame

In this paper, the CSIN model applies FPN for the target detection of construction personnel with high detection accuracy. The identity and location information of construction personnel can be obtained accurately in many cases. However, when shadows appear in the construction image, the accuracy of the CSIN model detection frame is affected to some extent. As shown in Figure 6d, the worker identification frame in the lower-left corner only detected the worker's head and hands, which may be due to the fact that the worker's legs blended into the shadow. Since FPN cannot distinguish shadows and targets better, it will affect the stability of the detection frame to a certain extent when shadows appear in the construction image. To solve this problem, generative adversarial networks (GAN) [37] or texture features of shadows in HSV space [38] for shadow suppression can be considered to eliminate the interference of shadows on image targets.

### 4.3. Generalization Capabilities of the CSIN Model

The CSIN model proposed in this paper was experimented on with self-made datasets. It has been verified that the model can complete target detection, visual relationship detection, and output construction scene information as text, realizing the automation and intelligence of identification in the key construction scenes on bridges. In the CSIN model, CNN and DCR are used as the base networks for target detection and relationship detection, respectively, which have been proved to have certain generalization abilities in related literature [39,40]. In addition, the underlying features of scene identification rules in this paper, such as color features, geometric features, and posture features, will not change greatly with different scenes, so they are portable. Therefore, the CSIN model can be applied to other types of construction and infrastructure projects, such as housing construction, road construction, etc. However, in port and tunnel construction, its generalization ability needs to be further verified due to the influence of datasets.

## 5. Conclusions

The construction process of highway bridges is tedious, and site environments are complex; thus, the realization of bridge construction scene identification helps relevant departments to carry out safety control. Therefore, based on the idea of visual relationships, this paper proposes the identification method of key construction scenes on highway bridges. This method can provide automated intelligent monitoring during the construction process and provide more applications for visual relationship detection in bridge construction. Firstly, the characteristics of bridge construction are analyzed and five key construction scenes are selected as research objects. Then, the scene identification rules are formulated from the three aspects of construction personnel, construction equipment, and construction materials. Following this, the CSIN model is built: FPN and color moments are first introduced to obtain the image features of construction workers, and solve the problem of missing and incorrect detection of target; then, through the division of subject–predicate–object triplet and image-text coding, the semantic features and visual features of construction scene can be obtained; finally, the integration features are matched with the

scene identification rules for consistency, and the category information of the construction scene is further obtained. Finally, the method in this paper is verified; the experimental results show that compared with other algorithms, the CSIN model obtained better results, especially on Recall@100.

Although the method in this paper has addressed the above problems, there are still two limitations. One is that the method is only experimentally validated in five key construction scenes, and research on other bridge construction scenes has not been carried out. The other is that the method involves fewer large equipment and construction materials, such as the lack of detection of large cranes, pile-driving machines, concrete, long bars, and other targets. Therefore, for the construction monitoring of different bridge types, such as girder bridges, arch bridges, rigid bridges, suspension bridges, cable-stayed bridges, and combined system bridges, it is necessary to further increase the identifiable elements in the construction scenes to enrich the bridge construction scene categories.

In our study, we found that the production of the dataset was time-consuming and laborious. In future work, we will combine efficient methods such as crowdsourcing labeling technology to produce targeted visual relationship detection datasets, so as to improve work efficiency. In addition, we will further optimize the CSIN model, combined with the relevant construction safety standards to realize the safety monitoring and safety assessment of bridge construction based on the existing methods. Thus, we will form a complete set of methods for intelligent monitoring and safety assessment of bridge construction, and extend it to other construction scenes.

**Supplementary Materials:** The following supporting information can be downloaded at: https://www.mdpi.com/article/10.3390/buildings12060827/s1.

**Author Contributions:** Conceptualization, C.W. and J.L.; Writing—Original Draft Preparation, C.W.; Writing—Review and Editing, C.W. and J.L.; data collection, Y.G.; data analysis, Y.L. All authors have read and agreed to the published version of the manuscript.

**Funding:** This research received no external funding.

**Institutional Review Board Statement:** Not applicable.

**Informed Consent Statement:** Not applicable.

**Data Availability Statement:** Some or all data are available from the corresponding author by request.

**Acknowledgments:** The authors are very thankful to those who made suggestions and comments to helpe improve the manuscript quality.

**Conflicts of Interest:** The authors declare that the research was conducted in the absence of any commercial or financial relationships that could be construed as a potential conflict of interest.

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
