# Peer review of "Visual Relationship-Based Identification of Key Construction Scenes on Highway Bridges"

_buildings, doi:10.3390/buildings12060827_

Round 1

Reviewer 1 Report

The paper addresses an important issue "Visual Relationship-Based Identification of Key Construction Scenes on Highway Bridges ". Overall, the paper is interesting from practical and theoretical point of view, as it provides target detection, visual relationship extraction, semantic conversion, scene information fusion and identification results output. However, the paper reveals some major deficiencies. The complements required for improvement of the paper are about:

  • Literature about object relationship detection through deep learning approach could be improved.
  • The purpose of the study could be better explained in introduction.
  • Research method could be presented in a flow-chart to better summarize the steps in the study.
  • Authors should explain clearly why Feature Pyramid Network (FPN) method was chosen for object detection.
  • Authors should explain why other algorithms, such as Large-Scale, Motifs, VRD, and Graph R-CNN have been selected for comparison of results.
  • Discussion is totally missing in the paper. Author(s) should discuss and elaborate more on the proposed CSIN model framework in terms of identifying key construction scenes of highway bridges based on visual relationships and mention any previous studies to allow comparison between the results obtained.
  • The potential implications in terms of generalization capabilities of the CSIN to other types of construction and infrastructure projects must be included and elaborated on.

Reviewer 2 Report

The submitted article is an interesting approach to a relevant investigation area and launches another approach to the consolidation of pattern recognition in civil engineering.

As a contextualization, there is a lack of further academic state-of-the-art similar methods.

I would recommend that figures 2 and 3 should be more readable as it is very difficult to understand what is written. My advice is to redesign them.

The Figure 2 legend should be succinct. All the detailed information about it should be described in the text. It is advised that the image source should be in the legend.

Authors refer to the "small portion of experimental data". Can the authors measure its statistical relevance? Otherwise how reliable are the results? How can they be compared to the other mentioned methods?

Discussion and conclusions should be increased and rewritten.

  • some edition comments:

The figure legends should be all in bold or all not in bold!

The legends should be on the same page as the respective image.

 On line 141 there's  a missing 1)

Reviewer 3 Report

This paper proposes a deep learning method for identifying key construction scenes of highway bridges based on visual relationships. This paper suggests that it would address the following problems:

  1. Provide an automated intelligence in safety monitoring during construction.
  2. Provide solution to the problem of missing and incorrect detection of targe
  3. Provide more application of visual relationship detection in highway bridge construction.
  4. Provide a classification of key construction scenes on bridges.
  5. Formulate identification rules for key construction scenes on bridges.
  6. Build an identification model for key construction scenarios on bridges.
  7. Validation of scene identification method. (This may not be stated in the introduction). The paper should explicitly show in the paper where all the above-listed problem has been addressed.

In addition, the following should be addressed.

  1. The conclusion of the paper cannot be in the induction. See line 73.
  2. The discussion section is underdeveloped.
  3. It is common to see the limitation of the research as the second part of the conclusion before the future direction of the research. Thus, the limitation of the research should be moved from the discussion section to the conclusion to maintain the typical convention.
  4. I am forced to read a lot of sentences more than once. This implies there is a need to improve the English language. MDPI English edit can help in improving the quality of the manuscript.

Round 2

Reviewer 1 Report

After revision, the originality, importance, value added of the paper and potential contribution to the journal has been improved. 

The revised paper has been well articulated dealing with sufficiently new and original concepts. Review of the various related literatures is comprehensive and provides an extensive and systematic mapping of object relationship detection through deep learning approach. The purpose of the study is clearer now. Flow-chart clearly summarizes the research method. The explanations regarding FPN method and other algorithms, such as Large-Scale, Motifs, VRD, and Graph R-CNN have been selected for comparison of results have been included. The authors have added a discussion part. The potential implications of CSIN to to other types of construction and infrastructure projects have been included.

Author Response

We would like to thank the referee and editor again for taking the time to review our manuscript.

Reviewer 2 Report

The authors took into consideration most of the reviewer´s comments and it´s clear the consequent improvement of the paper.

However, some remarks have to be done concerning the very long sentences and paragraphs that limit the understanding of the reader, as in lines 82-87 and  590-595.

Lines 231-242 should be aligned with the rest of the text.

Reviewer 3 Report

Congratulations.

Author Response

We would like to thank the referee and editor again for taking the time to review our manuscript.

This manuscript is a resubmission of an earlier submission. The following is a list of the peer review reports and author responses from that submission.